# Streamlining Analysis of RR Interval Variability in Elite Soccer Players: Preliminary Experience with a Composite Indicator of Cardiac Autonomic Regulation

**DOI:** 10.3390/ijerph17061844

**Published:** 2020-03-12

**Authors:** Daniela Lucini, Angelo Fallanca, Mara Malacarne, Maurizio Casasco, Leonarda Galiuto, Fabio Pigozzi, Giorgio Galanti, Massimo Pagani

**Affiliations:** 1BIOMETRA Department, University of Milano, 20133 Milano, Italy; angelo.fallanca@unimi.it (A.F.); mara.malacarne@unimi.it (M.M.); massimo.paganiz@gmail.com (M.P.); 2Exercise Medicine Unit, Humanitas Clinical and Research Center, 20089 Rozzano, Italy; 3FMSI Federazione Medico Sportivo Italiana, 00196 Rome, Italy; presidente@fmsi.it; 4Cardiology Department, Catholic University Sacred Heart, 0168 Rome, Italy; leonarda.galiuto@unicatt.it; 5Department of Movement, Human and Health Sciences, University of Rome “Foro Italico”, 0168 Rome, Italy; fabio.pigozzi@uniroma4.it; 6Clinical and Experimental Medicine Department, School of Sports Medicine, Sports Medicine and Exercise Center, University of Florence, 50141 Florence, Italy; giorgio.galanti@unifi.it

**Keywords:** soccer, cardiovascular regulation, autonomic nervous system, spectral analysis, endurance training, elite athletes

## Abstract

It is well recognized that regular physical activity may improve cardiac autonomic regulation preventing chronic non-communicable diseases. Accordingly, the assessment of cardiac autonomic regulation (CAR) with non-invasive techniques, such as RR interval Variability (V) might be of practical interest. We studied 56 soccer players (21.2 ± 4.2 years.) and 56 controls (22.2 ± 1.5 years.) and used a ranked Autonomic Nervous System Index (ANSI), resulting from the combination of multivariate statistical methodologies applied to spectral analysis derived indices from RRV. We hypothesized that ANSI would be higher in soccer players as compared to controls (*p* < 0.001) and that values would be greatest in defenders and midfielders, who are known to run longer distances during competitions. Conversely in the intrinsically stationary goalkeepers ANSI would be similar to controls. Our data show that it is possible to assess the overall level of autonomic performance in soccer players as compared to the general population, using a ranked composite autonomic proxy (ANSI). This approach suggests as well that CAR is better in those players who during competitions run for a greater distance. We conclude that it is possible to highlight the differences in autonomic profile due to distinct exercise routines, using ANSI, a simple ranked, composite autonomic proxy.

## 1. Introduction

Almost 300 million players, and their continuously rising number, render soccer the world’s number one sport [1], both in industrialized and low-income countries. Moreover, across Europe, the Americas, Middle East and Asia marketing investigations show that more than 40% of the population are interested in soccer (named football in Europe), corresponding to more than 700 million people [2]. This extraordinary reach gives soccer a unique position [3], facilitating the motivation to participate [4] to modern programs intended to reduce sedentariness and eventually improve cardiovascular risk profile [5]. Participation of children and adolescents in soccer activities may also be facilitated because it is considered as a source of enjoyment [6] and it is organized [7]. Accordingly playing soccer may counteract the decline of physical activity occurring towards adulthood [8]. Soccer may in addition be particularly successful in younger males [9] even if in these last years this sport has begun to be practiced by females too. Finally, compared with other sport activities, soccer has a highly dynamical nature [10,11]. Thus it may be well positioned to improve blood pressure, heart rate, glycometabolic profile and body composition as well as to increase aerobic fitness, since they all result from endurance training [12]. Among various mechanisms implicated in these improvements, Cardiac Autonomic Regulation (CAR), monitored through autonomic indices such as heart rate (HR) [13,14] and HR Variability (V) [15], may play an important role. In this context the protective cardiovascular effects of exercise may be greater than predicted by usual risk factors (lipids, hypertension, metabolism, etc.) and depend upon a favorable engagement of the autonomic nervous system resulting in a “risk factor gap” [16]. 

However, in spite of the presence of more than 25,000 hits in the Medline database, documenting the interest on the methodology, there are still several critical aspects in the study of this multidimensional phenomenon requiring clarification. To address part of these criticalities [17,18] we recently introduce a novel, composite, ranked Autonomic Nervous System Index (ANSI) [11,17,19,20]. ANSI is computed as a radar plot providing a synthesis of the three most informative spectral derived indices [RR Mean, RR variance (RR VAR), and the rest-stand difference in the normalized power of low-frequency (RR LFnu) variability component] of RRV, individually selected employing factor analysis, expressed as percent rank in view of the need of rendering the variables comparable [21]. ANSI, also percent ranked, and taken as combined proxy of CAR [22] is easier to compare across individuals or times, and would thus appear capable to provide a simple, convenient descriptor of CAR independent of age and gender [23,24]. 

Here we explore the hypothesis that the difference in CAR between a control population and a population of elite athletes could be easily demonstrated using ANSI. We hypothesized that the players of a national A series soccer team, taken as representative of long-term elite endurance sport participation, would demonstrate values of ANSI higher than observed in a similar age population of healthy sedentary controls. In view of the association between autonomic regulation and aerobic fitness [25] in addition we hypothesized that ANSI would be highest in midfielders and defenders who usually sustain the greatest load during competitions [26]. 

## 2. Materials and Methods 

### 2.1. Study Population and Protocol

This proof of concept, observational, retrospective study is part of an ongoing series of investigations, focusing on the use of autonomic indices in cardiovascular prevention, following a general protocol that had been approved by Independent Ethics Committee Humanitas Research Hospital (Rozzano, Italy) on 13 October 2015. Data refer to the entire group of a male soccer team of the Italian major league (series A), (*n* = 56; age 21.2 ± 4.2 years.) and to a similar age and gender (all male) group of healthy individuals, serving as control population (*n* = 56; age 22.2 ± 1.5 years.). This latter group is part of a continuously growing population of volunteers that usually visit our outpatient Exercise Medicine Clinic for reasons varying from a health check-up to cardiovascular prevention. The good health was ensured in athletes by their team doctor (following Italian law that prescribes annual preparticipation screening in competing athletes) or, in controls, by family physician (who provided information on normalcy of biochemical values such as glucose and lipids) and confirmed by history and physical examination. All subjects had provided informed consent at the time of the visit, they were informed and agreed that their anonymized data could be used for scientific projects. The protocol of this study followed the principles of the Declaration of Helsinki and Title 45, US Code of Federal Regulations, Part 46, Protection of Human Subjects, Revised 13 November 2001, effective 13 December 2001. 

### 2.2. Autonomic Evaluation

Our approach to the non-invasive evaluation of autonomic regulation has recently been summarized [22]. In brief, after an overnight fast and a light breakfast, avoiding caffeine and intense physical activity in the preceding 24 h, ECG and respiratory activity (piezoelectric belt, Marazza, Monza, Italy) are acquired on a PC. Beat-by-beat data series of 5 min rest followed by 5 min upright data are analyzed off-line with dedicated software (AMPS-llc, New York, NY, USA) [27]. As described previously [14], from the autoregressive spectral analysis of RR interval a series of indices indirectly reflecting cardiovascular autonomic modulation is derived using an ad hoc software requiring minimal operator involvement (Table 1).

The software tool [27] labels spectral components with a center frequency of 0.03–0.14 Hz as RR LFa, and components within the range 0.15–0.35 Hz as RR HFa, verifying the existence of an elevated coherence between RR variability and respiration (Figure 1). Recordings of subjects with arrhythmias or low frequency breathing are discarded in order to avoid bias which would render data uninterpretable [18]. Systolic and diastolic arterial pressure were measured using an electronic sphygmomanometer.

### 2.3. ANSI, a Proxy of Cardiac Autonomic Regulation (CAR)

ANSI was recently developed as a simple proxy of CAR resulting from the combination of the most informative *latent* factors hidden in the global set of variability data [19]. Factor analysis indicates that collectively three factors (related to domains of HR as well as amplitude and oscillatory RRV) account for about 80% of Variance Accounted For (VAF). According to loading values, they can be represented by the following indices: RR Mean, and RR VAR at rest, largely reflecting the vagal modulation, and the stand-rest difference in RR LFnu, as an index of the effects of the sympathetic excitation mediated by baroreceptor unloading. These variables are percent ranked and combined using a radar plot in a single proxy of CAR. The resulting ANSI easily indicates autonomic performance as a ranked percent where low and high values indicate poor and good performance of CAR, respectively.

### 2.4. Statistics

Data are presented as mean ± SD. Differences between controls and soccer players were assessed with an unpaired T-test for normally distributed variables, but with Mann-Whitney non parametric test for non-normally distributed variables (Shapiro Wilk test). Differences between groups (controls and soccer players according to their field position) were assessed with 1WAYANOVA followed by individual contrasts; normality of distribution was assessed with the Shapiro-Wilk test and for non-normally distributed data we employed the Kruskal Wallis analysis followed by Mann-Witney tests. *p* < 0.05 was set as the level of significance. Computations were performed using a commercial statistical package (SPSS version 24) (IBM, Armonk, NY, USA).

## 3. Results

As shown in Table 2, control subjects and A series soccer players (all male) were of similar age (22 vs. 21 years) and similar BMI (23 vs. 23 kg/m^2^), while arterial pressure was slightly elevated in soccer players yet still within normal range. Heart Rate was as expected lower in players (65 vs. 51 b/min). Players also displayed a lower value of double product, i.e., an indirect assessment of cardiac O_2_ consumption (7612.27 ± 1496.53 vs. 6583.67 ± 1223.06, b/min∙mmHg *p* < 0.001).

Regarding autonomic indices, soccer players, globally (Table 2), displayed higher RR VAR, and a rightward shift of spectral distribution (higher RR HFnu, together with lower RR LFnu and RR LF/HF). Concomitantly average ANSI was markedly higher in soccer players (see Figure 2).

Considering the specificity of player position (Figure 2 and Table 3) data suggest an uneven distribution of CAR. In detail, ANSI and other autonomic indicators in goalkeepers appear close to controls (the only significant difference with controls being a slightly longer RR interval); conversely defenders and midfielders show the greatest difference from controls; forwards’ profile is intermediate. Using ANSI as composite proxy of CAR, it is apparent that defenders and midfielders enjoy the best rest value, while goalkeepers are non-significantly different from normal controls.

## 4. Discussion

### 4.1. General Findings

Findings of this proof of concept, observational study, show that ANSI, a composite ranked proxy of CAR, is higher in players of an Italian Series A soccer team as compared to similarly aged controls. Moreover ANSI is highest in midfielders and defenders, who usually sustain the greatest external load during competitions. This approach might represent a convenient model to study the effects of long-term physical exercise on CAR. Moreover it is also suitable to distinguish the specificity of different playing positions.

### 4.2. Non-Invasive Assessment of Cardiac Autonomic Regulation (CAR)

Following the seminal study of Akselrod et al. in 1981 [28], cardiac autonomic regulation (CAR) in clinical applications is frequently gauged by proxies derived from analysis of beat by beat variations of cardiovascular activity, such as Heart Rate (HR) [29,30] or RR interval Variability [31,32,33,34,35] (V).

However, several critical aspects require clarification. The neural model underlying CAR can be either purely one-way motor as suggested by Langley [36] or bidirectional, afferent-efferent, networked multiple feed-back, unitary, cybernetic system, as originally proposed by WR Hess [37].

The methodology of analysis regards short vs. long term recordings [31]; time vs. frequency domain algorithms [31]; raw vs. mathematically manipulated indices, e.g., normalized units [14], and the origin of tachograms (from sinus sequences [18] of ECG derived RR intervals vs. PP intervals [38] or peripheral pulse [39]). Moreover capacity to extract information [40] on sympathetic-parasympathetic regulation or prevalence of either autonomic branch within a model of dynamic balance [14,41,42] should consider different meaning according to frequency codes for Low Frequency—LF—or High Frequency—HF—indices that seem invariant from periphery to central neural structures [43].

Statistical options vary from simple first order, monovariate descriptive or mix of robust, non-parametric and resampling techniques and application of exploratory statistical and graphical tools to ANS proxies [19] inclusive of simple or multivariate analysis, data transformation and aggregation [21].

Finally clinical applications are contingent upon attendant physiology such as age, gender, rest, exercise, sleep, or disease like hypertension, infarction, diabetes, etc; or treatments, for instance autonomic blockers e.g., [15].

To address part of these issues we recently introduced a novel, composite, ranked Autonomic Nervous System Index (ANSI) [11].

ANSI is obtained by integrating the information carried by three principal “latent” information determinants (RR Mean, RR VAR and stand-rest difference in RR LFnu), providing a unitary proxy of CAR, expressed as percentile rank against a reference benchmark population. The 0–100 ranking facilitates comparison between individuals, conditions or times [21]. ANSI is also built inherently non sensitive to age and gender. Finally percentiles are considered capable to convey a more immediate appreciation of CAR since they are standardized simply as 0–100 ranks, whereby higher is better. It is also important to reemphasize [40] that we are dealing with indirect data, hence variability proxies (e.g., LF component of RR variability) cannot provide detailed information of actual, raw electrophysiological figures of nerve activity but only suggest hypothesis about [44] general properties of autonomic balance [45].

An additional point in favor of the use of ANSI to assess CAR is represented by the strong correlation (r = 0.523, *p* < 0.001, see [19]) with the cardiac baroreflex gain, further supporting the interpretation that common mechanisms underlie both indicators.

From a practical clinical perspective, reducing a multidimensional phenomenon to a unitary, combined proxy of CAR, based only on single lead ECG recording, and 0-100 ranked, might further contribute to the introduction of autonomic evaluation in sports training. For instance it could be easier to follow the progressive changes in CAR occurring during an entire season or the possible variations in autonomic settings induced by competition stress.

### 4.3. Limitations and Perspectives

There are limitations to consider:

First ANSI as a proxy of CAR, like usual HRV derived autonomic indices, does not provide information on actual raw values of nerve activity (e.g., neural spikes as function of time) but only a value of ranking against a reference population.

In addition, ANSI does not address separately “frequency specific contributions” extracted from the tachogram but furnishes an integrated proxy of “the functioning of the principal cardiovascular control system: the sympathetic [and] parasympathetic systems” [28]. We feel however that ANSI would be “the appropriate technique of investigation” [37] to introduce evaluation of CAR also in sports other than soccer, because it is simple, economical, time efficient and immediate to understand thanks to the rank grading.

Future work should be focused on even simpler approaches to assess CAR, thanks to the emergence of ever more powerful wearable, miniaturized instruments, which could even combine GPS-based assessment of external load with indices of CAR, as a proxy of internal load.

## 5. Conclusions

The use of ANSI as a proxy of CAR permits one to unequivocally demonstrate a marked superiority of soccer players as compared to the control population, just using a single parameter. Moreover, considering different players’ position it is easy to recognize that those who are exposed to the greatest external load during competitions (i.e., midfielders and defenders) show the greatest values of ANSI (reaching almost 80%). Goalkeepers, who are exposed to the least external load, are just slightly superior to controls. We have shown that ANSI, i.e., a simple, unitary, index from RRV, can provide a proxy of CAR that can be unambiguously appreciated by its ranked value. We have tested ANSI’s worth as a simple index of autonomic performance in a sport of large interest worldwide. We document ANSI’s sensitivity as it discriminates between various players position. We hypothesize that ANSI could furnish a convenient method to assess CAR in other sports specialties [46], or in normal individuals undergoing long term physical training to achieve an improved CAR.

## Figures and Tables

**Figure 1 ijerph-17-01844-f001:**
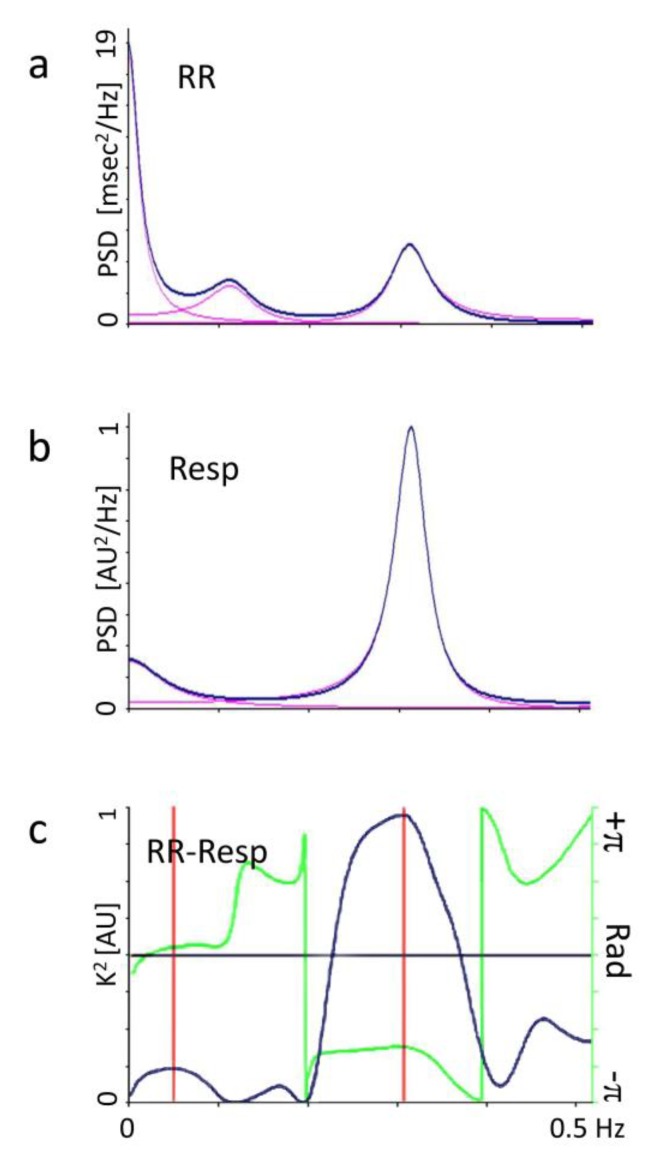
Panel a (labelled RR) depicts an example of an autospecrum (PSD) of RR interval variability obtained at rest on a study participant. The overall spectrum is drawn in blue, and as customary in healthy normal subjects, three major components are observed. They correspond to three individual spectral oscillations (indicated in red), that are recognized by the software tool. In normal conditions there is a component synchronous with respiration (denominated High Frequency, HF, about 0.25 Hz) and a second component at a Lower Frequency (LF, about 0.10 Hz). A third component around 0Hz corresponds to DC noise and very low frequency shifts of the RR interval signal (denominated usually VLF). Smaller noise components may also be present, usually below 5% of oscillatory power. Notice that ordinates should be multiplied by 1000. Panel b (labelled Resp) depicts an example of respiratory autospectrum. Notice that a single major component (at high, respiratory frequency) is observed typical of resting, physiological breathing. The Panel c (labelled RR-Resp) depicts in blue the coherence function (k2) between RR interval variability (i.e., tachogram) and respiration signal. In this optimal case it corresponds to almost 1. At any rate also smaller values, at least greater than 0.5 indicate, a significant exchange between RR interval and respiration; values smaller than 0.5 indicate non-significant coherence. In green is also depicted, for completeness, the Phase function (from −π to +π). This example depicts the well known phenomenon of the tachycardia accompanying inspiration (in this example the phase difference is about −0.6 rad corresponding to about 120°). Slow breathing (around 0.1 Hz) clouds the interpretation of RR interval autospectra. (see also 14, 18, 27).

**Figure 2 ijerph-17-01844-f002:**
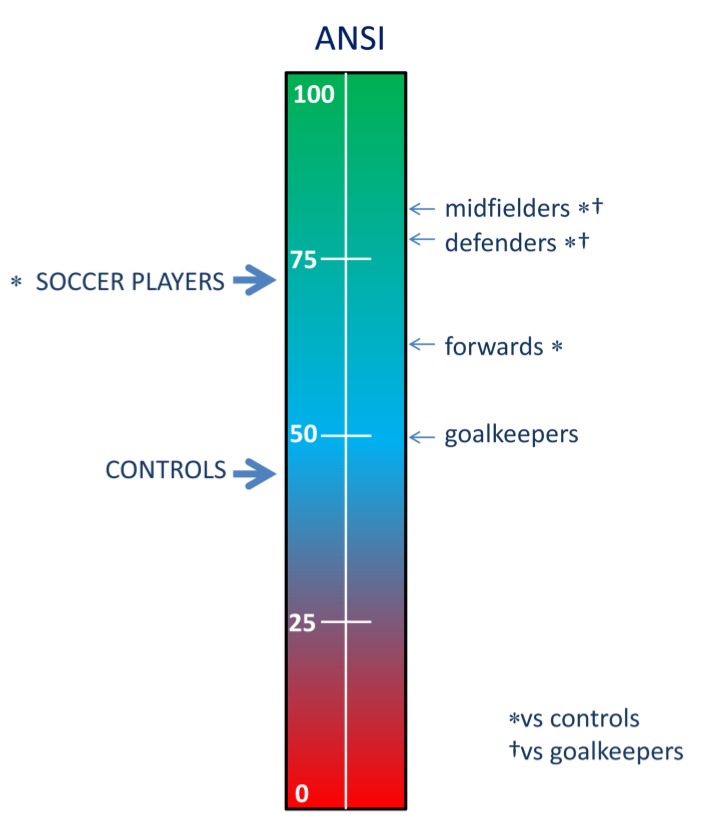
Different values of the unitary autonomic nervous system index (ANSI) in controls and Italian A team soccer players, according to playing position. * and † indicate significance versus respectively controls and goalkeepers.

**Table 1 ijerph-17-01844-t001:** Definition of the variables (ANS proxies) employed in the study ^(a)^.

Variables		Definition
HR	[beat/min]	Heart Rate
RR Mean	[msec]	Average of RR interval from tachogram sections
RR VAR	[msec ^2^]	RR variance from tachogram sections
RR LFa	[msec ^2^]	Absolute power(a) of Low Frequency (LF) component of RR variability (V)
RR HFa	[msec ^2^]	Absolute power(a) of High Frequency (HF) component of RRV
RR LFnu	[nu]	Normalized power (nu) of Low Frequency (LF) component of RRV
RR HFnu	[nu]	Normalized power (nu) of High Frequency (HF) component of RRV
RR LF/HF	.	Ratio between absolute values of LF and HF
ΔRRLFnu	[nu]	Difference of LF power in nu between stand and rest
SAP	[mmHg]	Systolic arterial pressure by sphygmomanometer
DAP	[mmHg]	Diastolic arterial pressure by sphygmomanometer
ANSI ^(b)^	[%]	Composite index of Autonomic Nervous System regulation computed as a synthesis of RR Mean, RR TP, and ΔRRLFnu

^(a)^ Modified from [22]. ^(b)^ Definition in [11].

**Table 2 ijerph-17-01844-t002:** Descriptive population values and cardiac autonomic indices in a group of Italian A series soccer players and a group of control subjects.

Variable		Controls *n* = 56	Soccer Players *n* = 56	*p*
**Age**	**[years]**	22.2 ± 1.5	21.2 ± 4.2	0.115
Weight	[kg]	74.2 ± 9.7	76.9 ± 6.0	0.240
Height	[cm]	178.7 ± 7.2	182.6 ± 5.7	< 0.001
BMI	[kg/m^2^]	23.2 ± 2.3	23.0 ± 1.0	0.443
SAP	[mmHg]	117.2 ± 10.5	130.2 ± 14.2	< 0.001
DAP	[mmHg]	68.4 ± 9.7	71.9 ± 7.3	0.011
HR	[beat/min]	65.1 ± 10.7	50.7 ± 8.4	< 0.001
RR	[msec]	946.3 ± 155.5	1213.2 ± 185.9	0.025
RR VAR	[msec^2^]	3970.4 ± 3504.2	10,729.2 ± 17,096.8	< 0.001
RR LFa	[msec^2^]	1189.7 ± 966.0	2072.4 ± 2630.1	0.249
RR HFa	[msec^2^]	1540.0 ± 2290.6	5530.5 ± 11,374.6	< 0.001
RR LFnu	[nu]	51.5 ± 21.8	32.3 ± 15.4	< 0.001
RR HFnu	[nu]	42.3 ± 21.2	64.9 ± 16.8	<0.001
RR LF/HF	.	2.43 ± 3.89	0.63 ± 0.57	<0.001
ANSI	[%]	45.5 ± 27.6	72.7 ± 23.7	<0.001

*p* = significance, by unpaired T-test, but with Mann-Whitney non parametric test for non-normally distributed variables (Shapiro Wilk test). Abbreviations: BMI = Body Mass Index. Other abbreviations: see Table 1.

**Table 3 ijerph-17-01844-t003:** Individual autonomic proxies according to playing position in an Italian Serie A soccer team.

Variable		Goalkeepers *n* = 5		Defenders *n* = 18		Midfielders *n* = 21		Forwards *n* = 12		*p*
**Age**	**[years]**	23.8 ± 5.9		20.5 ± 3.9	^‡^	21.6 ± 4.3		20.6 ± 3.9	^‡^	0.030
Weight	[kg]	84.6 ± 4.3	^‡^	79.4 ± 5.5	^‡^	73.9 ± 4.1	*+	75.2 ± 6.2	*	0.011
Height	[cm]	188.6 ± 2.1	^‡^	185.1 ± 4.8	^‡^	180.0 ± 4.5	*+	180.8 ± 6.6	*	< 0.001
BMI	[kg/m^2^]	23.8 ± 0.9		23.1 ± 0.9		22.8 ± 1.2		23.0 ± 1.1		0.719
SAP	[mmHg]	127 ± 10		130 ± 12	^‡^	133 ± 17	^‡^	126 ± 14		< 0.001
DAP	[mmHg]	71 ± 9		70 ± 8		73 ± 7		73 ± 7		0.091
HR	[beat/min]	55.2 ± 9.9		51.0 ± 9.9	^‡^	48.1 ± 6.5	^‡^	52.9 ± 8.1	^‡^	< 0.001
RR	[msec]	1118 ± 214.8	^‡^	1210.8 ± 196.5	^‡^	1268.2 ± 167.9	^‡^	1160.2 ± 179.4	^‡^	< 0.001
RR VAR	[msec^2^]	4192.4 ± 2027.2		10,625 ± 9932.3	^‡^	10,003.9 ± 10,352.8	^‡^	14,878.4 ± 32,763.6		0.003
RR LFa	[msec^2^]	1180.1 ± 618.5		1849.1 ± 2049.7		2019 ± 1908.2		2872.7 ± 4493.1		0.848
RR HFa	[msec^2^]	1554.6 ± 1042.9		5573.2 ± 6845.3	^‡^	5083.9 ± 7379.6	^‡^	7904.5 ± 21,479.4		< 0.001
RR LFnu	[nu]	43 ± 16.2		25.1 ± 13.4	^‡^*	31.1 ± 14.9	^‡^	40.8 ± 13.6	+	< 0.001
RR HFnu	[nu]	55.1 ± 16.9		71.1 ± 16	^‡^*	67.8 ± 15	^‡^	54.8 ± 16.3	^‡^+	< 0.001
RR LF/HF	.	1.02 ± 1.00		0.43 ± 0.43	^‡^*	0.53 ± 0.38	^‡^	0.92 ± 0.69	+	< 0.001
ANSI	[%]	50.1 ± 27.7		77 ± 22.9	^‡^*	79.6 ± 17.8	^‡^*	63.4 ± 26.4	(^‡^)	< 0.001

Data are expressed as mean ± standard deviation; *p* = overall significance by 1WAYANOVA, followed by individual contrasts. For non-normally distributed data we used Kruskal Wallis followed by individual contrasts with Mann-Whitney test. In this computation also the control population (data in Table 2) was considered. Significant (*p* < 0.05) individual contrasts: ^‡^ vs. controls; * vs. goalkeepers; + vs. defenders; # vs. midfielders. Abbreviations: BMI = Body Mass Index. Other abbreviations: see Table 1. Notice that the symbol for ANSI contrast between forwards and controls showing a marginal significance (*p* = 0.056) has been placed in brackets.

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
