# Peer review of "Streamlining Analysis of RR Interval Variability in Elite Soccer Players: Preliminary Experience with a Composite Indicator of Cardiac Autonomic Regulation"

_ijerph, 2020, doi:10.3390/ijerph17061844_

Round 1

Reviewer 1 Report

Review report of IJERPH-731343 by Lucini et al. Feb. 2020

GENERAL:
This is a clearly written manuscript. Some technical details can be omitted for the expected readership, especially if those are already published elsewhere.

MAJOR:
Concentrate in the DISCUSSION on the topic of this study, i.e. evaluation of soccer players.
Section 4.2 can be shortened, and written as a fluent text, rather than itemized.

MINOR:
line 42: <more that 40%> should be <more than 40%>
line 52: glicometabolic must be  glycometabolic
line 65: perhaps <need of rendering>
line 68: <observational, retrospective> also mentioned in line 77
line 79: add location to <Humanitas Research Hospital>
line 80: perhaps <series A>
line 85: <Italian law>, not <Italian low>
line 89: <statistical or scientific> why <or>? Do they exclude each other? I suggest to leave out <statistical or>
Figure 1. Label panels a, b, and c. Explain PSD in caption.
Line 127: perhaps better as <to their field position>
Line 136: include units
Table 3: include number of participants
Line 175 Akserlod should be Akselrod
line 200: perhaps as <we recently introduced>

Author Response

Cover 1

Please find itemized below the constructive Reviewer's questions and relative answers

Q1

GENERAL:

This is a clearly written manuscript. Some technical details can be omitted for the expected readership, especially if those are already published elsewhere.

A1

Thank you for the favorable comments. 

Regarding the technical details we had to balance it with the request of Reviewer 3 to better explain some specific aspects of the example of figure 1.  We hope that you will be satisfied by the present balance of space and details

Q2

MAJOR:

Concentrate in the DISCUSSION on the topic of this study, i.e. evaluation of soccer players.

Section 4.2 can be shortened, and written as a fluent text, rather than itemized.

A2

We agree on the importance of focusing just on the specific sport specialty (soccer).  Although one must concede that the same technique of autonomic analysis could be applied to different specialties, with potentially important translational effects (see e.g. ref 49).

The Section 4.2, was shortened from 624 to 512 words, and reorganized as fluent text, according to your suggestion .   

Q3

MINOR:

line 42: <more that 40%> should be <more than 40%>

line 52: glicometabolic must be  glycometabolic

line 65: perhaps <need of rendering>

line 68: <observational, retrospective> also mentioned in line 77

line 79: add location to <Humanitas Research Hospital>

line 80: perhaps <series A>

line 85: <Italian law>, not <Italian low>

line 89: <statistical or scientific> why <or>? Do they exclude each other? I suggest to leave out <statistical or>

Figure 1. Label panels a, b, and c. Explain PSD in caption.

Line 127: perhaps better as <to their field position>

Line 136: include units

Table 3: include number of participants

A3

Series of small changes all introduced

We thank you for the careful indication of specific minor points that could be improved.  We introduced all of them (L 42, L 52, L 65, L 179, L 80, L 85).

In particular, the point relative to L 68 and L 77, now reads:

-Here we explore the hypothesis…..

-This proof of concept, observational, retrospective study

And L (ex-89) 90 now reads 

….data could be used for scientific projects

Reviewer 2 Report

This work presents a very well elaborated study, which clearly extracts and exposes very interesting results for training and elite football. I think that with the quality of this work it should be accepted for publication.

Author Response

Cover 2

Q1

This work presents a very well elaborated study, which clearly extracts and exposes very interesting results for training and elite football. I think that with the quality of this work it should be accepted for publication.

A1

We sincerely thank this Reviewer for his/her very kind and favorable comments.

Reviewer 3 Report

In this manuscript the authors described in their preliminary study that Autonomic Nervous System Index (ANSI) is capable to provide a simple, convenient descriptor of cardiac autonomic regulation independent of age and gender.

Furthermore, the authors in their preliminary study demonstrated how the players from elite endurance sport participation have values of ANSI higher than in similar population of healthy sedentary controls. In addition, the authors demonstrated that ANSI is higher in midfielders and defenders who usually sustain the greatest load during competitions.

By that results ANSI could serve a convenient method to assess CAR in other sports specialties as well.

However, some changes should be made in this paper for further consideration.

Since this is a preliminary study, which involves a low number of participants, in limitation part of the study should be stated that this study should be confirmed on a lager number of participants.

In addition, in methods part, participants should be better described, the exact number of defenders, and midfielders, goalkeepers, etc including the Table 3 (n=?)

Figure 1. The figure must be improved with better labeled ordinates and panels (a,b and c); resolution is to low and the text in hard to read or not at all. The figure should be described better and all the parts in it, to be clearly understandable by a wider readership, especially panel c.

Author Response

Cover 3

Please find below itemized  Reviewer 3 thoughtful questions and relative answers

Q1

GENERAL

In this manuscript the authors described in their preliminary study that Autonomic Nervous System Index (ANSI) is capable to provide a simple, convenient descriptor of cardiac autonomic regulation independent of age and gender.

Furthermore, the authors in their preliminary study demonstrated how the players from elite endurance sport participation have values of ANSI higher than in similar population of healthy sedentary controls. In addition, the authors demonstrated that ANSI is higher in midfielders and defenders who usually sustain the greatest load during competitions.

By that results ANSI could serve a convenient method to assess CAR in other sports specialties as well.

However, some changes should be made in this paper for further consideration.

A1

We thank this Reviewer for his/her thoughtful general comments.  In particular he/she wrote that  “ANSI could serve a convenient method to assess CAR in other sports specialties as well”.

He/she also indicated that some changes to the submitted text were appropriate.  These changes are itemized below

Q2

Since this is a preliminary study, which involves a low number of participants, in limitation part of the study should be stated that this study should be confirmed on a larger number of participants.

In addition, in methods part, participants should be better described, the exact number of defenders, and midfielders, goalkeepers, etc including the Table 3 (n=?)

Figure 1. The figure must be improved with better labeled ordinates and panels (a,b and c); resolution is to low and the text in hard to read or not at all. The figure should be described better and all the parts in it, to be clearly understandable by a wider readership, especially panel c.

A2

The following sentence has been added (L 241):

Finally, since this is a preliminary study, which involves a low number of participants, results should be confirmed on a larger number of subjects.

The following sentence has been added to the Materials and Methods Section,

L 82  “Before enlistment the project was explained to the prospective participants by one of the Authors (DL).  We enrolled all players on a voluntary basis, treating  everybody (goal keepers, defenders, midfielders and forwards) as equal.”

The number of players for any subgroup is now indicated in table 2.

Thank you for this last important comment.  We tried our best to accommodate your suggestion, with the concern for length of methods expressed by Reviewer 1.  Figure 1 has been modified in order to improve its overall quality.

Here we report the new legend for Figure 1

L 113

Panel a  (labelled RR) depicts an example of an autospecrum (PSD) of  RR interval variability obtained at rest on a study participant.  The overall spectrum is drawn in blue, and as customary in healthy normal subjects, three major components are observed.  They correspond to three individual spectral oscillations (indicated in red), that are recognized by the software tool.  In normal conditions there is a component synchronous with respiration (denominated High Frequency, HF, about 0.25 Hz) and a second component at a Lower Frequency (LF, about 0.10 Hz).  A third component around 0Hz corresponds to DC noise and very low frequency  shifts of the RR interval signal (denominated usually VLF).  Smaller noise components may also be present, usually below 5% of oscillatory power.  Notice that ordinates should be multiplied by 1000.  Panel b (labelled Resp) depicts an example of respiratory autospectra. Notice that a single major component (at high, respiratory frequency is observed) typical of resting, physiological breathing. The Panel c (labelled RR-Resp) depicts in blue the coherence function (k2) between RR interval variability (i.e. tachogram) and respiration signal.   In this optimal case it corresponds to almost 1.  At any rate also smaller values, at least greater than 0.5 indicate, a significant exchange between RR interval and respiration; values smaller than 0.5 indicate non-significant coherence.  In green is also depicted, for completeness,  the Phase function (from –  to + ) showing.  This example depicts the well known phenomenon of the tachycardia accompanying inspiration (in this example the phase difference is about -0.6 rad corresponding to about 120°). Slow breathing (around 0.1Hz) clouds the interpretation of RR interval autospectra. (see also [14, 18, 27])

Round 2

Reviewer 3 Report

The authors improved their paper very well and I have no further comments. However, I didn't see the  following sentences authors stated they have been inserted them into line 82 and line 241:

L 241:

"Finally, since this is a preliminary study, which involves a low number of participants, results should be confirmed on a larger number of subjects."

L 82:

“Before enlistment the project was explained to the prospective participants by one of the Authors (DL). We enrolled all players on a voluntary basis, treating everybody (goal keepers, defenders, midfielders and forwards) as equal.”

I hope that will be made after.